# *Bacillus thuringiensis* Cry14A family proteins as novel anthelmintics against gastrointestinal nematode parasites

Duy Hoang[1☉], Kelly Flanagan[1☉], Qian Ding[1‡], Nicholas R. Cazeault[1‡], Hanchen Li[1‡], Stefani Díaz-Valerio[2], Florentina Rus[1], Esther A. Darfour[1], Elizabeth Kass[3], Katherine H. Petersson[3], Martin K. Nielsen[4], Heiko Liesegang[2], Gary R. Ostroff[1], Raffi V. Aroian[1]*

**1** Program in Molecular Medicine, UMASS Chan Medical School, Worcester, Massachusetts, United States of America, **2** Department of Genomic and Applied Microbiology & Göttingen Genomics Laboratory, Institute of Microbiology and Genetics, Georg-August University of Göttingen, Göttingen, Germany, **3** Department of Fisheries, Animal, and Veterinary Sciences, University of Rhode Island, Kingston, Rhode Island, United States of America, **4** M.H. Gluck Equine Research Center, Department of Veterinary Science, University of Kentucky, Lexington, Kentucky, United States of America

☉ These authors contributed equally to this work.
‡ QD, NRC and HL also contributed equally to this work.
* raffi.aroian@umassmed.edu

**Data Availability Statement:** All the crystal proteins used in this research can be found at Genbank: Cry14Ab1 has accession no. KC156652

## Abstract

*Bacillus thuringiensis* crystal (Cry) proteins have been expressed in commercial transgenic crops for nearly 30 years, providing safe and effective control of insect pests and significantly reducing the application of hazardous chemical pesticides. *B. thuringiensis* crystal proteins have also been shown to target parasitic nematodes, including plant parasitic nematodes. Recently, transgenic soybean crops expressing Cry14Ab have been shown to provide control against the soybean cyst nematode *Heterodera glycines*, marking the first time a crystal protein is being commercialized in transgenic crops for control of a nematode pest. However, apart from *H. glycines* and the free-living nematode, *Caenorhabditis elegans*, the breadth of nematode activity of Cry14Ab, *e.g.*, against gastrointestinal parasitic nematodes (GINs), has not been reported. Here we study the efficacy of Cry14Ab against a wide range of gastrointestinal nematode parasites (GINs) in vitro and in vivo. We find that Cry14Ab is effective in vitro against the barber's pole worm *Haemonchus contortus* larvae, small strongyles cyathostomin larvae, the hookworm *Ancylostoma ceylanicum* adults, the roundworm *Ascaris suum* L4 larvae, and the whipworm *Trichuris muris* adults. In rodents infected with GIN parasites, Cry14Ab is effective as an in vivo anthelmintic against the hookworms *A. ceylanicum* and *N. americanus*, against the mouse parasite *Heligmosomoides polygyrus bakeri*, and against the roundworm *A. suum*. Cry14Ab also variably reduces the reproduction of the whipworm *T. muris* in vivo. Using optimized profile Markov Models, we looked for other putative anthelmintic Cry proteins and, within this list, identified a Bt crystal protein, GenBank accession no. MF893203, that we produced and demonstrated intoxicated GINs. This protein, with 90% amino acid identity to Cry14Ab, is active against *C. elegans*, *A. ceylanicum* adults, and *A. suum* L4 larvae in vitro. MF893203 was given the official designation of Cry14Ac. Cry14Ac is also an effective in vivo anthelmintic against *A. ceylanicum*

and Cry14Ac has accession no. MF893203. All other relevant information are in the manuscript and its Supporting information files.

**Funding:** This work was financially supported by the National Institutes of Health National Institute of Allergy and Infectious Diseases grants R01-AI056189 to R.V.A. and USDA-NIFA-AFRI Grant no. 2021-67015-34574 from the USDA National Institute of Food and Agriculture to K.H.P. The funders had no role in study design, data collection and analysis, decision to publish, or preparation of the manuscript.

**Competing interests:** I have read the journal's policy and the authors of this manuscript have the following competing interests: The University of Massachusetts Chan Medical School is pursuing patent protection related to the use of Cry14Ab as anthelmintic via U.S. Provisional Patent Application Serial No. 63/439,759 (K.F. and R.V.A. as inventors) and the University of Massachusetts Chan Medical School and University of Göttingen are pursing patent protection related to the use of Cry14Ac as anthelmintic via U.S. Provisional Patent Application Serial No. 63/649,526 (K.F., S.D.V., H. L., and R.V.A. as inventors).

hookworms in hamsters and intestinal *A. suum* in mice. Taken together, our results demonstrate that Cry14Ab and Cry14Ac have wide therapeutic utility against GINs.

## Author summary

Gastrointestinal parasitic nematodes or worms pose a significant threat to global health, causing illness in millions-billions of people and animals. Current treatments have limitations, including concerns about drug resistance. We are exploring a promising new approach using natural Crystal (Cry) proteins from *Bacillus thuringiensis* (Bt) bacteria, which are already widely used as safe and effective insecticides. To date, we have focused on the anthelmintic (deworming) properties of Bt Cry protein Cry5Ba. Here, our research demonstrates that a newly characterized and different Bt Cry protein, Cry14Ab, can also effectively target and kill various types of harmful parasitic nematodes in both laboratory and animal models. Moreover, we identify another related Bt protein, Cry14Ac, with similar activities against parasitic nematodes. These findings highlight the potential of Bt proteins as a novel class of deworming medications, offering hope for the development and clinical deployment of safer and more effective treatments against parasitic nematode infections.

## Introduction

*Bacillus thuringiensis* or Bt is a common gram-positive soil bacterium that produces pesticidal proteins [1]. The best studied are three-domain crystalline (Cry) proteins that accumulate in parasporal crystal inclusions during sporulation [1]. Each Cry protein kills only a narrow set of target invertebrates, including major insect pests [2]. Cry proteins are biodegradable and safe to all vertebrates, including humans, as well as to most beneficial invertebrates [3]. As such, Cry proteins have been used for decades to kill insect vectors of disease as well as insect crop pests in conventional and organic farming [1], accounting for ~90% of biopesticides sold worldwide [4]. Cry proteins have been produced in transgenic crops such as corn, cotton, and soybean that were planted on a cumulative total of 1.5 billion hectares from 1996 to 2022 [5,6]. More than a dozen Cry proteins produced by transgenic crops have been studied and approved as safe for human consumption by the US Environmental Protection Agency (EPA) and Food and Drug Administration (FDA) [7,8]. Hundreds of Cry proteins in >50 different families have been identified and characterized [9,10].

Although commercial use of Bt and Bt Cry proteins to date has only applied to insect pests, studies have suggested Bt Cry proteins, closely related to those that target insects by sequence and structure [11–14], are also relevant for controlling nematode pests, the most prevalent and ubiquitous eukaryotic parasites on earth. With regards to human and animal pests, orally-delivered Cry5Ba protein has been shown to be therapeutically effective in mammalian hosts against a wide array of gastrointestinal nematode (GIN) parasites, including hookworms, ascarids, barber's pole worm, and *Heligmosomoides polygyrus bakeri* [15–24]. With regards to plant pests, transgenic tomato roots expressing the Cry protein Cry5Ba showed protection against the root-knot nematode *Meloidogyne incognito*, inhibiting the establishment of infection sites and reducing progeny production 3-fold [25]. This latter use of a nematode-active Cry protein is the closest to the current use of Cry proteins against insect pests.

Based on this result, the related Cry protein, Cry14Ab1 (hereafter Cry14Ab), was explored for control of *Heterodera glycines*, the soybean cyst nematode, one of the most important disease agents of soybeans worldwide [26] (Cry14Ab falls broadly into the phylogenetic family of Cry proteins that target nematodes such as Cry5B [14,27]). Transgenic soybean plants expressing Cry14Ab significantly impaired the reproduction of *H. glycines* in both greenhouse and field trials [27]. These transgenic Cry14Ab soybean plants have received regulatory approval from the US EPA and FDA for deployment against *H. glycines* [28,29], suggesting they may be released soon as a commercial trait.

Apart from *H. glycines*, the activity of Cry14Ab against other nematode pests has not been reported. Indeed, although the related protein Cry14Aa has been identified as having activity against free-living nematodes or stages of nematodes [14,30,31], to date no Cry14A protein family member has been demonstrated to have anthelmintic activity, *e.g.*, therapeutic activity against GIN parasitic infections [32,33]. Because of the approval of Cry14Ab in a food crop, we set out to examine if Cry14Ab has anthelmintic properties as well. Here, we test Cry14Ab against GIN parasites in vitro and in vivo. Using optimized profile Markov Models to identify novel anthelmintic family proteins in sequence databases such as UniProt, we also uncover a new anthelmintic member of this family, now called Cry14Ac1 (or abbreviated here as Cry14Ac) and explore its anthelmintic properties.

## Results

### Cry14Ab can be produced as IBaCC

We recently reported on a novel, safe, and scalable method for production of the nematode-active Cry5Ba proteins for use as an anthelmintic, called IBaCC for *In*activated *Ba*cteria with *C*ytosolic *C*rystal(s) [20,22,24]. We cloned Cry14Ab downstream of the *cry3Aa* promoter to express it in vegetative cells and then transformed the construct into *Bacillus thuringiensis* 4D8 cells in which we had deleted the *spo0A* gene to prevent sporulation. As with Cry5Ba, Cry14Ab was robustly produced in vegetative cells in Δ*spo0A* cells under control of the *cry3Aa* promoter (Fig 1A and 1B). These cells were then inactivated (killed) with essential oil treatment as described [24], resulting in no live cells (Fig 1C).

### Cry14Ab is effective against GINs in vitro

Cry14Ab IBaCC was then quantitated against bovine serum albumin by SDS-PAGE and used in dose-response bioassays against the larval stages of two important veterinary GIN parasites, the blood-feeding parasite of sheep, *Haemonchus contortus*, and a common parasite of horses, cyathostomins, small strongyles that encompass 40 species and that commonly occur as coinfections of 15–25 species [34]. Cry14Ab IBaCC is highly effective against the larval stages of *H. contortus* (Fig 2A) and cyathostomins (Fig 2B), with complete inhibition of development at 5 ng/mL (about 10X more potent than Cry5Ba IBaCC at 56 ng/mL or ~400 pM; [22]). Given that Cry14Ab is a 132 kilodalton (kDa) protein, this 5 ng/mL dose represents a concentration of 38 pM.

We tested Cry14Ab IBaCC in vitro against the parasite stages of GINs that infect or are closely related to those that infect humans. Adult *A. ceylanicum* parasites, which are prominent hookworm parasites of humans especially in Southeast Asia [35], were harvested from infected hamsters and treated with varying doses of Cry14Ab IBaCC in 48-well format. Twenty-four hours later, the motility of the parasites was measured as a proxy for nematode health/intoxication using the Worminator system set up in our laboratory [21]. As shown, the motility of the hookworms was significantly inhibited in a dose-dependent manner (Fig 3A).

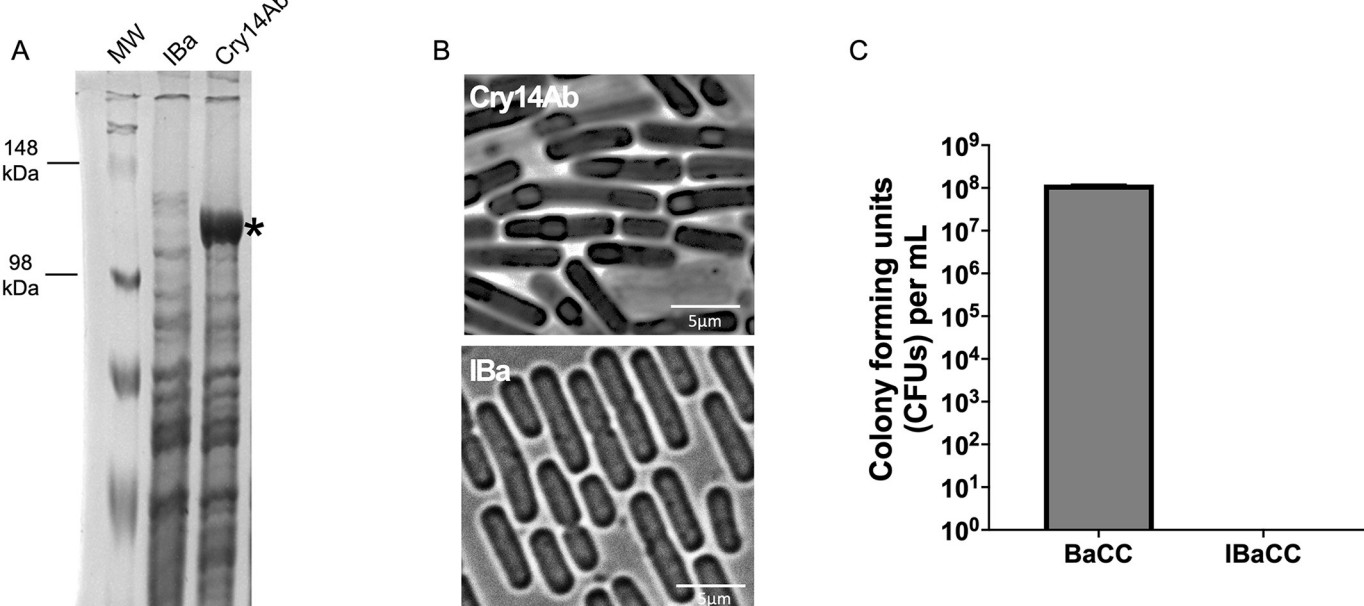

**Fig 1. Cry14Ab IBaCC.** A. Gel showing expression of Cry14Ab (*) in Bt 4D8 Δ*spo0A* cells. MW = molecular weight markers; IBa = inactivated bacteria with empty vector; Cry14Ab = Cry14Ab IBaCC. B. Photomicrograph of IBaCC cells expressing Cry14Ab or IBa cells (inactivated bacteria with empty vector). C. Colony forming units (CFUs) of Cry14Ab cells before (BaCC = **Ba**cteria with **C**ytosolic **C**rystals) and after (IBaCC) treatment with monoterpene. CFU/mL BaCC = 1.2 x $10^8$; IBaCC = 0 (average of three experiments).

We also tested lysed Cry14Ab IBaCC against intestinal L4 staged *Ascaris suum* (pig roundworm) parasites. We recently reported on a new system that allows for robust development of *A. suum* parasites in mice past the lung phase and into the intestinal L4 stage [20,21]. *A. suum* is a very close relative of, if not the same species as, *Ascaris lumbricoides* (human roundworm) [36–38]. We harvested fourth staged (L4) parasite *A. suum* larvae from the intestines of infected mice and studied their motility in 96-well format over time using the touch 3–0 scale

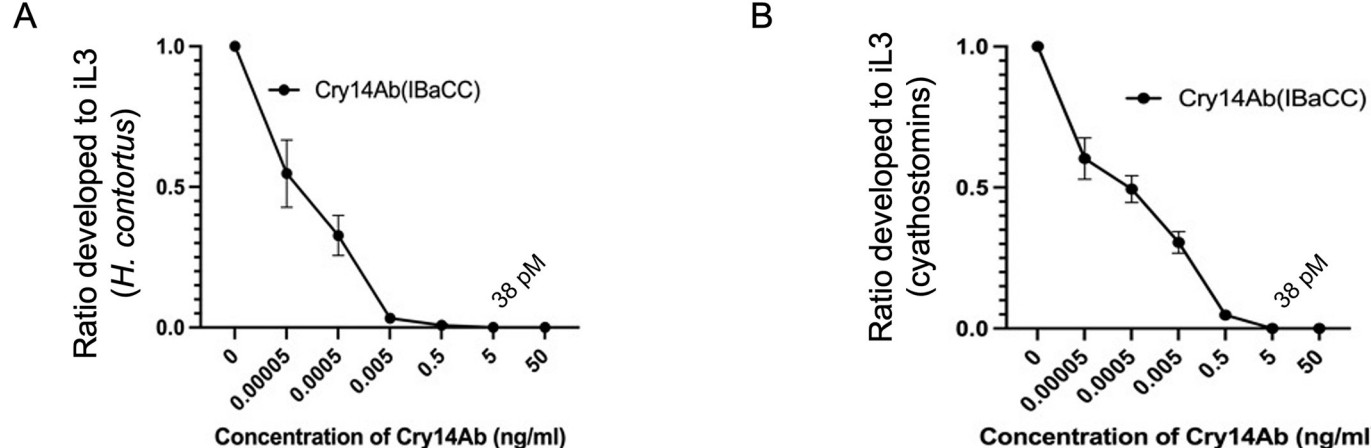

**Fig 2. Efficacy of Cry14Ab IBaCC against larval stages of veterinary GIN parasites.** A. Ratio of *Haemonchus contortus* L1 larvae that developed to the infectious third larval (iL3) stage on increasing doses of Cry14Ab (as IBaCC) relative to the number that developed to the iL3 in assay media without Cry14Ab (normalized to 1.0). B. Percent cyathostomin L1 larvae that developed to the iL3 stage on increasing doses of Cry14Ab (as IBaCC) relative to assay media without Cry14Ab (normalized to 1.0). n = ~ 50 L1s/well, repeated 3 independent times. Plotted are the mean and standard error.

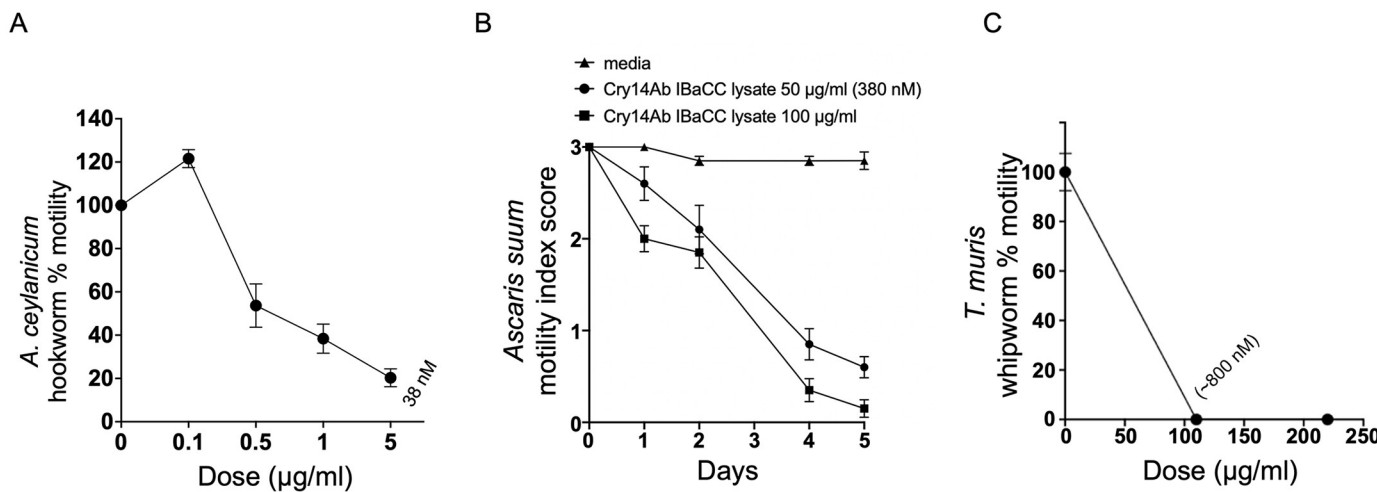

**Fig 3. Efficacy of Cry14Ab IBaCC lysate against parasitic stages of the three major human GINs.** (A) Dose-response relative motility assay of Cry14Ab IBaCC lysate against *Ancylostoma ceylanicum* adult hookworms (average of 3 experiments; n = 8/experiment). Here and in (C), motility was normalized to movement of parasites in the absence of Cry protein (buffer only) using the Worminator. (B) Motility of *Ascaris suum* intestinal L4 stage roundworms at two different doses of Cry14Ab IBaCC lysate over time. 3 = fully motile; 2 = inhibited motility; 1 = immotile until touched; 0 = immotile even with touch (n = 20 per dose; 5 per experiments repeated 4 times). (C) Motility of *Trichuris muris* whipworms in vitro relative to no Cry14Ab control measured using the Worminator 72 hours after exposure to Cry14Ab IBaCC lysate. Average of two repeats, n = 13–14 whipworms total.

([39]; this stage of *A. suum* is too small to be read with our Worminator). Cry14Ab IBaCC was also highly effective against this parasite in vitro (Fig 3B).

A third major human GIN is whipworm. We tested lysed Cry14Ab IBaCC in vitro against this parasite, in particular mouse whipworm or *Trichuris muris*. As determined by our Worminator system [21], Cry14Ab IBaCC completely inhibited the motility of whipworm adults at 72 hr. at a dose of ~800 nM (110 μg/mL; Figs 3C and S1).

## Cry14Ab is an effective anthelmintic in vivo

Given the in vitro activity of Cry14Ab against GINs, we next tested the in vivo anthelmintic activity of Cry14Ab to cure parasitic infections in rodents. Cry14Ab was given as a single oral dose to hamsters infected with either *A. ceylanicum* (Fig 4A) or *N. americanus* (Fig 4B) hookworms, which together encompass the two hookworm genera that infect humans. In both cases, significant reductions in parasite burdens and reproduction (fecal egg counts) were seen in Cry14Ab IBaCC treated groups relative to control groups. For example, single dose 20 mg/kg Cry14Ab reduced *A. ceylanicum* hookworm burdens by 68% and fecal egg counts by 92%. Single dose 50 mg/kg nearly eliminated *N. americanus* burdens. The doses used in Fig 4 would, at a molar level, not show any effects using other anthelmintics, *e.g.*, albendazole.

We also looked at the impact of Cry14Ab on luminal feeding GINs. A double oral dose of Cry14Ab IBaCC was given to mice infected with *A. suum* that progressed to the intestinal phase. As shown (Fig 5A), a near complete (95%) elimination of intestinal *Ascaris* burdens relative to control was seen. Similarly, a single oral dose of Cry14Ab IBaCC of mice infected with *Heligmosomoides polygyrus bakeri* led to significant reductions in intestinal worm burdens and parasite reproduction (fecal egg counts) (Fig 5B).

As we saw effects on the whipworms in vitro, we tested whether Cry14Ab would have impacts on whipworm infections in vivo. Since *T. muris* is present in the cecum and since we currently lack a formulation to protect Cry14Ab through the stomach and small intestine, we

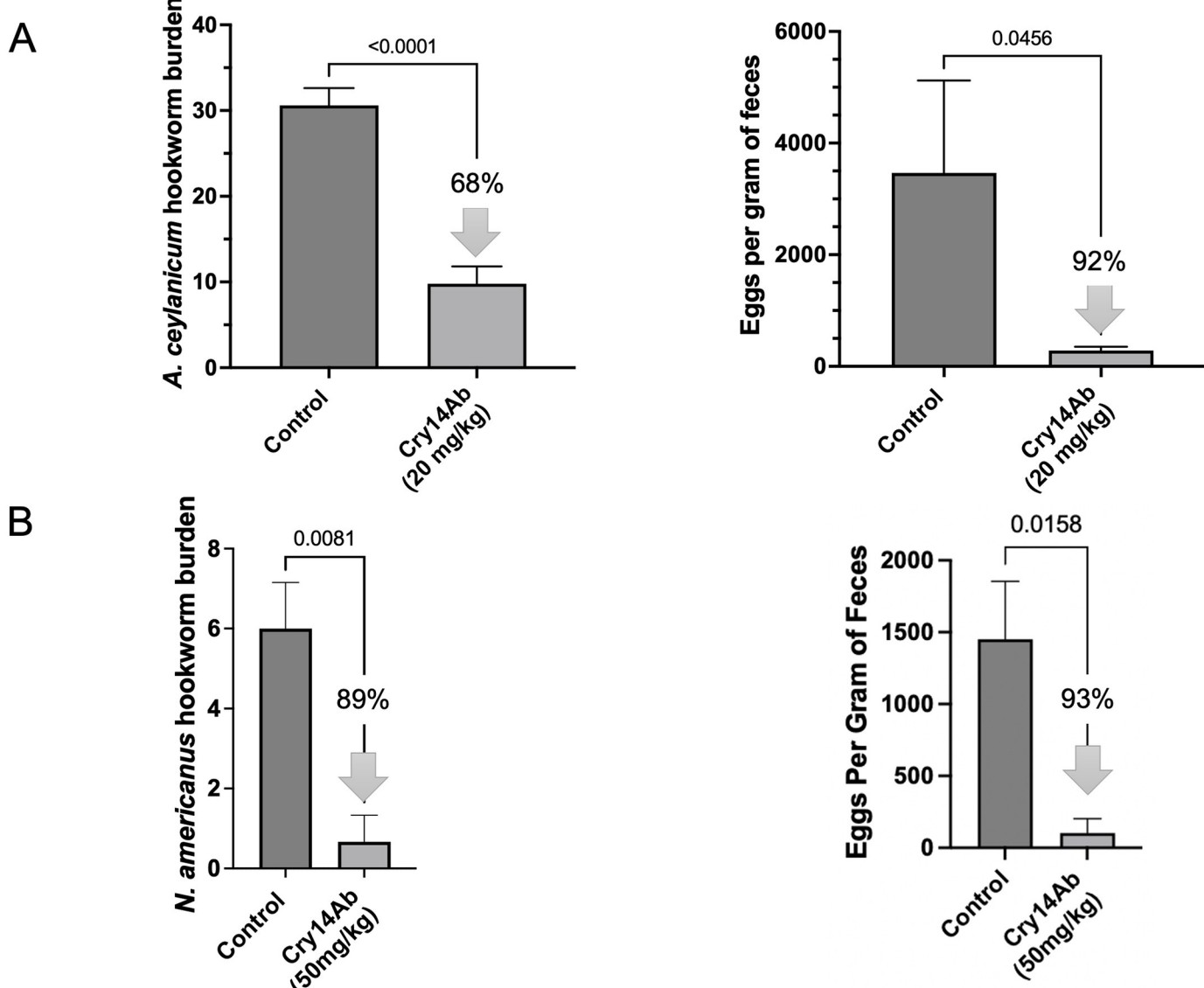

**Fig 4. Efficacy of Cry14Ab IBaCC against human hookworm infections in hamsters.** (A) *Ancylostoma ceylanicum* infections in hamsters in control (water) and Cry14Ab IBaCC treated groups. 20 mg/kg Cry14Ab is equivalent to 40 μg/kg albendazole on a molar basis. Left: Average intestinal hookworm burden per hamster; right: average fecal egg count per hamster (parasite reproduction). N = 5 hamsters/group. (B) *Necator americanus* infections in hamsters in control (water) and Cry14Ab IBaCC treated groups. 50 mg/kg Cry14Ab is equivalent to 100 μg/kg albendazole on a molar basis. Left: average intestinal hookworm burden per hamster; right: average fecal egg count per hamster (parasite reproduction). N = 3 hamsters/group. P values based on one-tailed Student's t test.

injected solubilized Cry14Ab protein directly into the small intestine of mice infected with *T. muris* and followed fecal egg counts. The experiment was repeated three times. In two of three experiments, Cry14Ab treatment resulted in a dramatic reduction in whipworm reproduction (S2 Fig).

## The newly uncovered and related protein Cry14Ac is also anthelmintic

The Liesegang laboratory has reported on an *in silico* approach for detecting and characterizing novel pesticidal proteins from publicly available sequence databases based on optimized

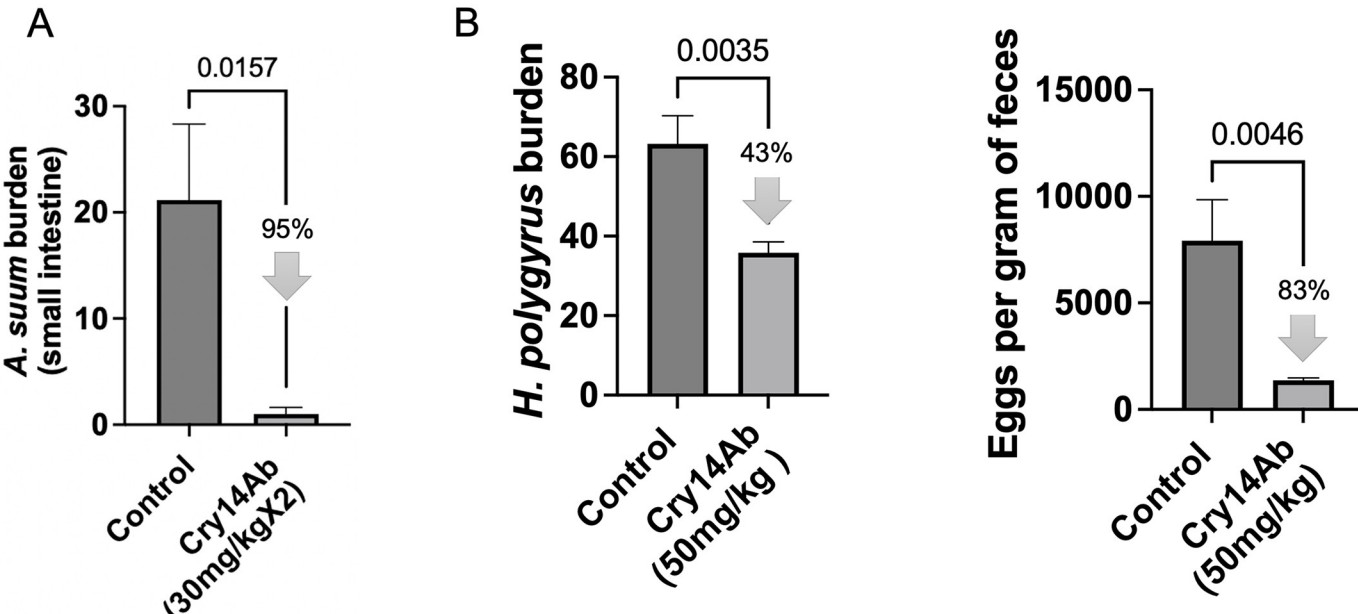

**Fig 5. Efficacy of Cry14Ab IBaCC against luminal feeding parasites.** (A) Average *Ascaris suum* intestinal worm burden per mouse in control (water; n = 6) and Cry14Ab IBaCC (n = 5) treated groups. Treated group: 30 mg/kg Cry14Ab (equivalent to 60 µg/kg albendazole on a molar basis) was given two days in a row to infected mice. (B) *Heligmosomoides polygyrus* infections in mice in control (water) and Cry14Ab IBaCC treated groups. 50 mg/kg Cry14Ab is equivalent to 100 µg/kg albendazole on a molar basis. Left: Average intestinal worm burden per mouse; right: fecal egg counts (parasite reproduction). N = 5 mice/group. P values based on one-tailed Student's t test.

profiling using hidden Markov models (HMM) methodology [40]. In the case of Cry proteins, the group is represented by several CRY-HMMs to account for the sequence diversity and modular nature of the protein class. The search was narrowed down based on HMM scores and domain signatures similar to those of known nematicidal Cry proteins according the IDOPS tool. This HMM tailored to recognize nematicidal Cry proteins was used on UniProt, giving rise to a number of putative anthelmintic Cry proteins. These were synthesized as genes and expressed. Among these, we found that the protein encoded by GenBank accession no. MF893203 was highly active against GINs (see below). MF893202 has 90% amino acid identity with Cry14Ab (S3 Fig), with 87% amino acid identity in the bioactive domain (crystal protein domains 1+2+3) and 94% amino acid identity in the crystallization domain (C terminal domain). MF893203 was previously identified in a Bt strain that also contained a Cry21Aa-family protein, but the biological activity of MF893203 was unknown at that time, and whether or not it had anti-nematode activity was not determined [41]. As such, MF893203 did not receive an official Cry protein designation [41]. As demonstrated below, M893203 is an anthelmintic and has now received the official designation of Cry14Ac1 (hereafter Cry14Ac).

We generated Cry14Ac IBaCC cells [24] (Fig 6A). To test for anti-nematode activity, we exposed fourth larval stage (L4) *C. elegans*, both wild-type and *bre-5(ye17)* mutants that are resistant to Cry5Ba [31,42], to Cry14Ac IBaCC. We found that Cry14Ac IBaCC is highly active against *C. elegans* and that, unlike Cry5Ba, is able to qualitatively intoxicate the *bre-5(ye17)* resistance mutant (Fig 6B), which lacks the functional receptor for Cry5Ba [43].

To test if Cry14Ac is also active against GINs, we used Cry14Ac IBaCC to treat in vitro parasitic intestinal stages of hookworms (*A. ceylanicum*) and roundworms (*A. suum*). Cry14Ac IBaCC was highly effective against both parasites in vitro (Fig 7).

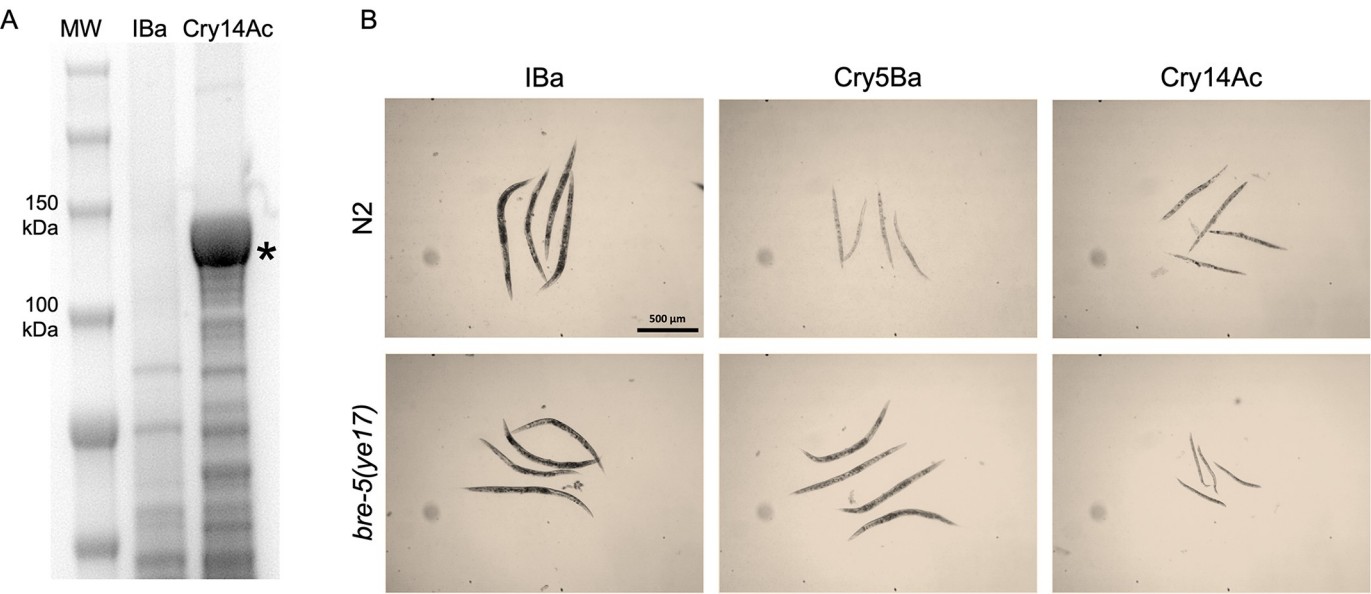

**Fig 6. Efficacy of Cry14Ac IBaCC against *Caenorhabditis elegans* nematodes.** A. SDS-PAGE showing expression of Cry14Ac (*) in IBaCC. Here and panel B: IBa = inactivated bacteria with empty vector; Cry14Ac = Cry14Ac IBaCC. MW = molecular weight markers. B. L4 *C. elegans* were grown in the presence of indicated bacteria for 6 days at 25° C. All images were taken at the same magnification. N2 = wild-type *C. elegans*. *bre-5(ye17)* = Cry5Ba-resistant *C. elegans*. Cry5Ba = Cry5Ba IBaCC. N2 animals are healthy (large, motile, and well-fed or dark) when fed IBa but are severely intoxicated (stunted, immotile, pale) when fed Cry5Ba or Cry14Ab IBaCC. Cry5Ba-resistant *bre-5(ye17) C. elegans* are healthy when fed IBa or Cry5Ba (towards which *bre-5(ye17)* but are intoxicated by Cry14Ac.

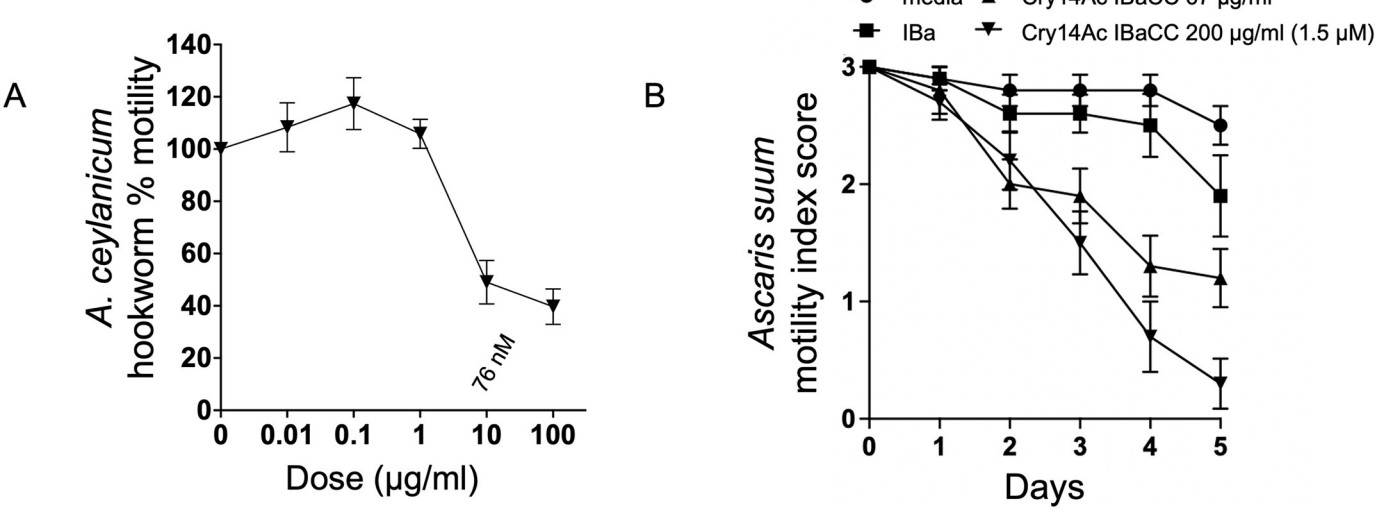

**Fig 7. Efficacy of Cry14Ac IBaCC lysate against parasitic stages of the two major human GINs.** (A) Dose-response relative motility assay of Cry14Ac IBaCC lysate against *Ancylostoma ceylanicum* adult hookworms (average of 3 experiments; n = 8/experiment). Motility is normalized to movement of hookworms in the absence of Cry protein using the Worminator. (B) Motility of *Ascaris suum* intestinal L4 stage at two different doses of Cry14Ac IBaCC lysate over time. 3 = fully motile; 2 = inhibited motility; 1 = immotile until touched; 0 = immotile even with touch (average of 2 experiments; n = 5 per dose per experiment). IBa = inactivated bacteria with empty vector.

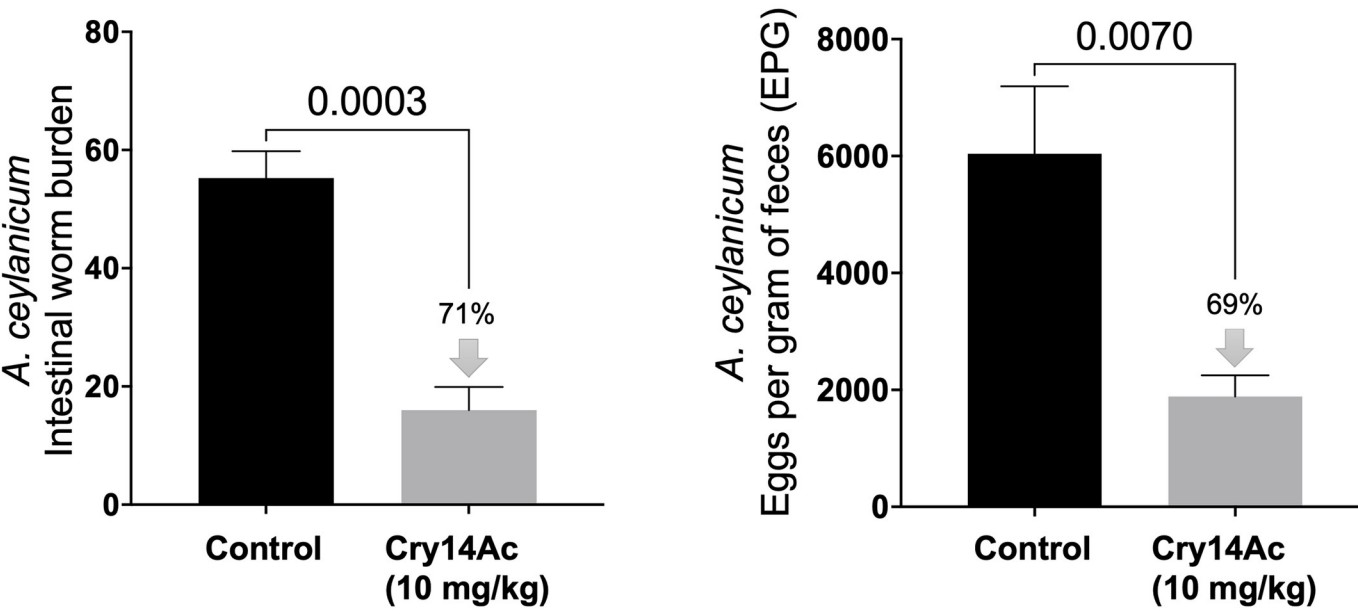

**Fig 8. Efficacy of Cry14Ac IBaCC against human hookworm infections in hamsters.** *Ancylostoma ceylanicum* infections in hamsters in control (water) and Cry14Ac IBaCC treated groups. 10 mg/kg Cry14Ab is equivalent to 20 µg/kg albendazole on a molar basis. Left: average intestinal hookworm burden per hamster; right: average fecal egg count per hamster (parasite reproduction). N = 4 hamsters/group. P values based on one-tailed Student's t test.

We also tested the anthelmintic activity of Cry14Ac in vivo against *A. ceylanicum* hookworm infections in hamsters and intestinal *A. suum* infections in mice. Single dose Cry14Ac (10 mg/kg) as IBaCC resulted in ~70% elimination of hookworm burdens and fecal egg counts (Fig 8A and 8B). Double dose Cry14Ac (20 mg/kg) as IBaCC resulted in strong (80%) reduction in *A. suum* intestinal burdens (Fig 9).

## Discussion

Here for the first time we demonstrate that Cry14A family proteins target human and animal parasitic nematodes. This work was motivated by the recent expression and regulatory approval of Cry14Ab expression in soybeans for transgenic control of the soybean cyst nematode. It was not known whether this protein, useful against plant-parasitic nematodes when expressed in transgenic plants, would have good efficacy against GINs.

Indeed, we find Cry14Ab protein is highly effective against the environmental larval stages of two important parasites in veterinary medicine, *H. contortus* (ovine host) and cyathostomins (equine host). Cry14Ab is also effective in vitro against intestinal parasitic stages of GINs in genera that encompass the three major parasites of humans: hookworms (*Ancylostoma*), roundworms (*Ascaris*), and whipworms (*Trichuris*). On a molar level, the GIN parasites are extremely sensitive to this protein in vitro, showing strong intoxication in the 40–800 nM range.

In parallel with results in vitro, we demonstrated that Cry14Ab is also an effective anthelmintic in vivo. Cry14Ab IBaCC is active against infections of two human hookworm species, *A. ceylanicum* and *N. americanus*, in hamsters. The efficacy is significant and high, albeit lower than that of our published Cry5Ba IBaCC data against the same hookworm infections (although the exact same doses were not used [24]). Cry14Ab is also a highly effective

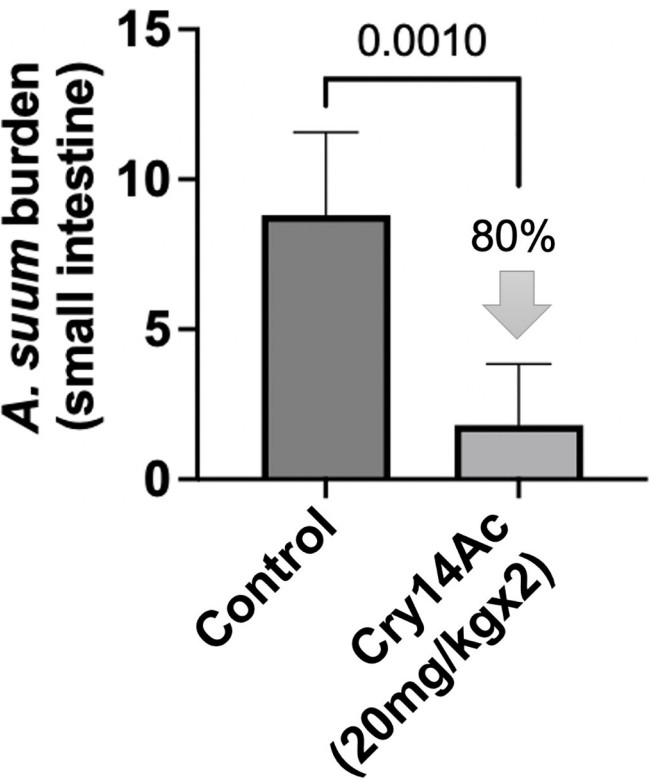

**Fig 9. Efficacy of Cry14Ac IBaCC against intestinal *Ascaris suum* infections in mice.** Average *A. suum* intestinal worm burdens per mouse in control (water; n = 5) and Cry14Ac IBaCC treated groups (n = 5). Treated group: 20 mg/ kg Cry14Ac was given two days in a row to infected mice. P values based on one-tailed Student's t test.

anthelmintic against luminal GINs, namely *A. suum* and *H. polygyrus bakeri* infections in mice, with efficacy similar to that published for Cry5Ba [20,24]. For the first time, we also report the variable efficacy of a Cry protein against *T. muris* whipworms, with Cry14Ab leading to significant reductions in fecal egg counts when injected directly into the intestine in two of three experiments. These data suggest potential for further optimization of this protein against whipworms, *e.g.*, via improved formulation and delivery.

Using a novel *in silico* method to detect and cluster pesticidal proteins, we also detected a new anthelmintic member of the Cry14A family, Cry14Ac. Cry14Ac readily overcomes a *C. elegans* strain resistant to Cry5Ba, indicating that it must, to some extent, use a different receptor than the *bre-5*-associated Cry5Ba glycosphingolipid receptor [43]. Moreover, Cry14Ac is effective in vitro against *A. ceylanicum* hookworm adults and *A. suum* L4 larvae. Cry14Ac is also an effective anthelmintic in hamsters infected with *A. ceylanicum* hookworms and mice infected with intestinal *A. suum* roundworms. In the case of hookworms, Cry14Ac efficacy appears to be superior to that of Cry14Ab.

An interesting question to consider is whether Cry14Ab expressed in a transgenic soybean plant can have any anthelmintic effect, *e.g.*, on GIN-infected farm animals. Although the levels of Cry14Ab expression in various tissues in these plants is not reported, Cry proteins can be expressed at levels ranging from 0.003–0.5% of total soluble protein in the plant [44,45]. Based on these data, we believe that the level of expression would not be sufficient to deliver an effective anthelmintic dose in normal feed.

This investigation also indicates that deep data mining techniques *in silico* are useful for uncovering new nematode-active Cry proteins. This study also demonstrates that the Cry14A family of proteins can broadly play an impactful role in the control and treatment of GIN parasites. Additional anthelmintics are critically needed as parasitic recalcitrance to synthetic anthelmintics continues to increase among human, livestock, and companion animal populations [24,46,47]. This report is also the first instance to demonstrate that multiple members of a single family of Cry proteins are anthelmintics and only the second, after Cry5Ba, to demonstrate that Bt Cry proteins have anthelmintic activity in vivo against a range of parasitic GINs. The ability of Cry14Ac to overcome Cry5Ba resistance suggests that multiple Cry protein families (*e.g.*, Cry5B, Cry14A) may be productively stacked (combined) to prolong their anthelmintic efficacy, as occurs in natural Bt strains and has been engineered into transgenic plants [48–50].

Given the global burden of GIN infections, the pressing need for new therapeutic options, and the increasing threat of anthelmintic resistance, this research represents a promising advancement in the field. The broad-spectrum activity of Cry14Ab, combined with its safety profile as part of approved transgenic crops and along with Cry14Ac, make these proteins promising candidates for further development as a novel anthelmintics.

## Materials and methods

### Microbiology methods

*Escherichia coli* 5α cells (NEB) were used for cloning and maintained on Luria-Bertani (LB) medium including ampicillin (100 μg/ml) at 37˚ C or erythromycin (300 ug/ml) at 30˚ C. *E. coli dam⁻/dcm⁻* (NEB) cells were used to generate nonmethylated plasmids for Bt transformations and similarly maintained on LB medium with appropriate antibiotics. *Bt* strains were maintained at 30˚ C with LB medium, including erythromycin (10ug/ml) for plasmid maintenance when appropriate. Solid media contained 1.6% agar. The Bt strains used here are Bt HD1 4D8, into which we generated a *spo0A* deletion (HD1 4D8 Δ*spo0A*) as part of another study (manuscript in preparation) and Bt 407 Δ*spo0A*::*kan* [24,51].

To express Cry14Ab1, called Cry14Ab throughout this manuscript, DNA encoding Cry14Ab1 (GenBank accession no. KC156652) was transcriptionally fused between the *cry3A* promoter region -635 to +18 [52] and the *cry5B* terminator in the shuttle vector pHT3101 [53], by Genscript (Piscataway, NJ), as previously done for *cry5B* expression [24]. Later, DNA encoding Cry14Ac (GenBank accession no. MF893203) was synthesized and cloned by Genscript into the expression cassette of pHY170 (aka pKF45 [54]), a pHT3101-derived plasmid which contains the promoter region of *cry3A* -635 to ATG and the downstream terminator of *cry5B*. Nonmethylated plasmids were sequenced again prior to use. For *cry14Ab*, electrocompetent HD1 4D8 Δ*spo0A* cells and 407 Δ*spo0A*::*kan* cells were subsequently transformed with the empty vector pHT3101 or the above *cry14Ab* construct (pHY171, P$_{cry3A}$-*cry14Ab*, aka pKF36). For *cry14Ac*, electrocompetent 407 Δ*spo0A*::*kan* cells were subsequently transformed with the empty vector pHT3101 or the above *cry14Ac* construct (pHY172, P$_{cry3A}$-*cry14Ac*, aka pDH13). Bt isolates harboring pHT3101 (empty vector), pHY171 (P$_{cry3A}$-*cry14Ab*), or pHY172 (P$_{cry3A}$-*cry14Ac*) were confirmed by PCR.

### Production and characterization of IBa and IBaCC

To produce any IBaCC, a single vegetative colony of Cry5Ba (Cry5Ba1), Cry14Ab-, or Cry14Ac-expressing asporagenous Bt was incubated in 3X LB at 25˚C with shaking at 125rpm for 3 days. Cry14Ab-expressing Δ*spo0A Bt* was similarly incubated in a 350L fermenter at a biomanufacturing facility with constant monitoring and adjustments of pH and oxygen levels

at 25˚C with 150 rpm agitation for 48 h. Exhausted expression cultures were harvested by centrifugation and resuspended to 10% initial volumes in ice-cold water, then processed to IBa (**I**nactivated **Ba**cteria with empty vector, *i.e.*, no Cry protein) or IBaCC as previously described [24]. Briefly, 10X concentrated cell suspensions were treated with 0.1% terpene for 15 minutes at room temperature and washed three times with ice-cold water and centrifugation. Lab-cultured IBa and IBaCC samples were additionally subjected to organic extraction with food-grade corn oil prior to water washes, to remove residual terpene. Samples of concentrated IBa/IBaCC suspensions were used to analyze Cry protein quantification, IBaCC cell viability, and assays for nematicidal bioactivity. The IBa (inactivated bacteria with empty vector) and IBaCC cells were observed using an Olympus BX60 microscope equipped with a UPlanFl 100x/1.30 Oil Ph3 objective (Fig 1B).

Cry protein content of IBaCC preparations were quantified by SDS-PAGE relative to Bovine Serum Albumin (BSA; Sigma-Aldrich) standards analyzed by densitometry using ImageJ software [55]. Cell and BSA samples were denatured in Laemmli sample buffer for denaturing SDS-PAGE using cast or precast acrylamide gels (Bio-Rad) with tris-glycine running buffer.

For the experiment shown in Fig 1C, two cultures of pHY171 (P$_{cry3A}$-*cry14Ab*) were grown in LB overnight at 30˚C in 10 μg/ml erythromycin till saturation. The next day, 100 μl of each culture was inoculated in 3X LB at 30˚C with 10 μg/ml erythromycin and agitated at 250 rpm for three days. Exhausted cultures were harvested and one was processed to IBaCC with our standard protocol (see above); the remaining culture was concentrated down 10-fold to match the IBaCC sample. After each sample was appropriately diluted, 100 μl was plated onto a LB plate and incubated at 30˚C overnight. Growth or lack of growth was recorded the following day. The data represent the average of three independent experiments.

To make IBaCC lysate (lysed IBaCC), IBaCC was lysed by vortexing in 50 mM Tris-HCl pH6.5, 250 mM NaCl, mutanolysin (35 U/mL), lysozyme (20 mg/mL), and 0.02% Pluronic F127. This suspension was rotated overnight at room temperature and then centrifuged at 5000xg for 20 min at 4˚C to pellet crystals, which were washed 3X in cold sterile water and stored in the same.

## Animal experiments

**Caenorhabditis elegans.** *Caenorhabditis elegans* was maintained using standard techniques [56]. The following strains were used in this study: N2 Bristol (wild type) and *bre-5 (ye17)* [42]. For images shown in Fig 6B, 20–40 L4 hermaphrodites were incubated in 160 μl S-medium supplemented with 20 μl *Escherichia coli* OP50 (OD600 ~ 3) and either 1) 20 μl S-medium, or (2) 20 μl IBa (inactivated bacteria with empty vector) suspended in S-medium or (3) 20 μl Cry protein IBaCC (final Cry protein concentration 50 μg/mL per well) suspended in S-medium in 48-well format. The images were taken after six days of incubation at 25˚C. Pictures are representative from two independent trials.

## Parasites and hosts

*Ancylostoma ceylanicum* and *Necator americanus* hookworm life cycles were maintained in hamsters as previously described [18] with the exception that *N. americanus* was maintained with use of 2 mg/liter dexamethasone in the drinking water. *H. polygyrus bakeri* life cycle was maintained in Swiss Webster mouse as previous described [18]. *Trichuris muris* life cycle was maintained in B6/STAT6KO mice as previously described [39]. *Ascaris suum* infectious-staged eggs were from Joseph F. Urban, Jr. at United States Department of Agriculture and shipped to University of Massachusetts Medical School. All rodent experiments were carried out under protocols approved by the University of Massachusetts Chan Medical School (IACUC;

PROTO202000044, PROTO202000071). The procedures for collection of fecal samples from sheep were approved by University of Rhode Island's Institutional Animal Care and Use Committee under protocol AN2021-013. The procedures for collection of fecal samples from horses were approved by University of Kentucky's Institutional Animal Care and Use Committee under protocol 2021–3879. All housing and care of laboratory animals used in this study conform to the National Institutes of Health (N.I.H., U.S.A.) Guide for the Care and Use of Laboratory Animals in Research (see 18-F22) and all requirements and all regulations issued by the United States Department of Agriculture (U.S.D.A.), including regulations implementing the Animal Welfare Act (P.L. 89–544) as amended (see 18-F23).

## In vitro experiments

Larval development assays for *H. contortus* and cyathostomins (Fig 2) were carried out as described [17,22]. Reagents for hookworm culture medium (HCM), including RPMI 1640 medium, fetal bovine serum (FBS), penicillin-streptomycin, and fungizone antimycotic, were all purchased from Gibco, USA. Adult *A. ceylanicum* hookworms harvested from the intestines of infected hamsters day 20 post-inoculation (PI) were assayed in vitro using the Worminator system to measure motility as previously reported [21] with the following modifications: 25 mM Hepes pH 7.2, 96-well format, 100 µL hookworm medium per well. For the Cry14Ab experiment, the AMU for 100% motility was 141; for the Cry14Ac experiment, 137.

 *A. suum* L4 parasites (day 12 harvested from the intestines of B6/STAT6KO mice; [20]) were exposed to no protein, 50 µg/ml, and 100 µg/mL Cry proteins for 5 days (37 ˚C, 5% $CO_2$) in RPMI 1640 with (all final concentrations) 25 mM HEPES (pH 7.2), 5% fetal bovine serum, anti-microbials (100 U/mL penicillin, 100 mg/mL streptomycin; 2.5 µg/mL amphotericin B). One adult parasite per well were placed in 100 µl medium in a 96-well format with the designated treatment using 5 wells/condition (mixed gender) and then set up 4 independent times for Cry14Ab (lysed IBaCC) and 2 independent times for Cry14Ac (IBaCC). *A. suum* L4s were scored on a 0 to 3 scale (0, nonmotile even when touched; 1, nonmotile unless touched; 2, slowly motile; 3, fully motile) as described previously [23]. The L4 stage (instead of the adult stage) is used for *A. suum* as STAT6KO mice expel this parasite after they reach the L4 intestinal stage but before they reach the adult intestinal stage.

 Adult *T. muris* in vitro assays were carried out using one adult per well (3–6 male and 3–6 female worms per condition) harvested between days 35–40 PI and placed in buffer (same as for *A. suum* above) in a volume of 300 µL in 48-well plates. IBaCC lysate was used for these studies as intact IBaCC is not accessible to whipworms. Whipworm motility was measured using the Worminator system set up in our laboratory [21]. The AMU for 100% motility was 100.

## In vivo experiments

The *A. ceylanicum* and *N. americanus* in vivo experiments were carried out as described previously [18,19,24]. For curative Cry14Ab IBaCC experiments in hamsters against *A. ceylanicum* only, cimetidine was given *per os* 15 min ahead of Cry5B administration, as previously described [18]. Experiments using *H. polygyrus bakeri* were carried out as described previously [24]. To determine the in vivo efficacy of Cry14Ab IBaCC and Cry14Ac IBaCC against intestinal *A. suum*, experiments were carried out as described previously [20], except B6/STAT6KO mice were inoculated *per os* with 5000 infective *A. suum* eggs. Treatment occurred on days 12 and 13 PI and intestinal burdens determined on day 18 PI. Intact IBaCC was used for all these in vivo experiments.

 For the *T. muris* in vivo experiments, the Cry14Ab was solubilized to make it accessible to the whipworm parasites. IBaCC lysate (see above) was solubilized in 20 mM citrate buffer pH

3 [57] that contained 0.1% Pluronic F-127 (Sigma-Aldrich, P2443). The solubilization process was carried out at 4 ˚C, with periodic mixing during a 30-minute incubation. After solubilization, the tubes were centrifuged for 10 minutes at 3500g. The resulting supernatant containing the soluble Cry14Ab protein was then transferred to an Amicon 50 kDa centrifugal filter (Millipore-Sigma, UFC905008) and concentrated. The concentrated soluble Cry14Ab protein was then immediately used for in vivo experiments.

To get pre-treatment parasite egg counts, feces were collected from individual mice on day 35 PI. Mice were treated with 500 μL of pH 3 solubilized Cry14Ab (50 mg/kg) on day 36 PI via direct injection into the distal small intestine of the mice to bypass the stomach. Control animals received 500 μL citrate buffer alone via direct instillation. Feces were collected overnight day 42–43 PI for fecal egg counts and then mice sacrificed for whipworm burdens [39]. For all in vivo experiments, worm burdens per host animal are shown.

### Statistics

All graphs were produced and statistical analyses (one-tailed Student's t test) carried out using GraphPad Prism v10. All the data used in plots are included in S1 Table.

## Supporting information

**S1 Fig. *Trichuris muris* in vitro motility experiment with Cry14Ab IBaCC compared to IBa (inactivated bacteria with empty vector).** We have already demonstrated that IBa has no effect on hookworms [24] and roundworms (Fig 7B). Two independent batches of Cry14Ab IBaCC were prepared. All conditions were repeated for a total of two times and n = 12 parasites per condition.
(TIF)

**S2 Fig. Efficacy of solubilized Cry14Ab IBaCC lysate against *Trichuris muris* whipworms in vivo.** Fecal egg counts were taken seven days after direct instillation of solubilized Cry14Ab (single dose, 50 mg/kg) in 20 mM citrate buffer into the distal small intestine of the mice. P values based on one-tailed Student's t test. Control groups received 20 mM citrate buffer (same volume). Three independent trials were carried out. In none of the experiments was a significant difference in whipworm burdens seen but in 2/3 experiments, a significant change in fecal egg counts was seen. Shown are the fecal egg counts and standard error for each group in each of the three experiments. For the experiment on the left, n = 5 control, n = 6 Cry14Ab; for the experiment in the middle, n = 4 for both groups; for the experiment on the right, n = 3 for both groups.
(TIF)

**S3 Fig. Alignment of full length Cry14Ab and Cry14Ac.** Identical residues are highlighted in green, conservative changes in yellow. Pairwise alignment was generated with Geneious Prime 2022.1.1.
(TIF)

**S1 Table. Means and standard error for all data points shown in graphs.**
(XLSX)

## Author Contributions

**Conceptualization:** Kelly Flanagan, Heiko Liesegang, Gary R. Ostroff, Raffi V. Aroian.

**Formal analysis:** Duy Hoang, Kelly Flanagan, Qian Ding, Nicholas R. Cazeault, Hanchen Li, Stefani Díaz-Valerio, Heiko Liesegang, Raffi V. Aroian.

**Funding acquisition:** Katherine H. Petersson, Raffi V. Aroian.

**Investigation:** Duy Hoang, Kelly Flanagan, Qian Ding, Nicholas R. Cazeault, Hanchen Li, Stefani Díaz-Valerio, Florentina Rus, Esther A. Darfour, Heiko Liesegang.

**Project administration:** Raffi V. Aroian.

**Resources:** Elizabeth Kass, Katherine H. Petersson, Martin K. Nielsen.

**Software:** Stefani Díaz-Valerio.

**Supervision:** Kelly Flanagan, Heiko Liesegang, Gary R. Ostroff, Raffi V. Aroian.

**Validation:** Florentina Rus.

**Visualization:** Duy Hoang, Kelly Flanagan, Qian Ding, Nicholas R. Cazeault, Hanchen Li.

**Writing – original draft:** Raffi V. Aroian.

**Writing – review & editing:** Kelly Flanagan, Gary R. Ostroff.

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
