## [Decision Letter · Decision Letter 0]

5 Jul 2024

Dear Prof. Aroian,

Thank you very much for submitting your manuscript "Bacillus thuringiensis Cry14A family proteins as novel anthelmintics against gastrointestinal nematode parasites" for consideration at PLOS Neglected Tropical Diseases. As with all papers reviewed by the journal, your manuscript was reviewed by members of the editorial board and by several independent reviewers. The reviewers appreciated the attention to an important topic. Based on the reviews, we are likely to accept this manuscript for publication, providing that you modify the manuscript according to the review recommendations. 

Sincerely,

David Joseph Diemert, M.D.

Academic Editor

jong-Yil Chai

Section Editor

Reviewer's Responses to Questions

**Key Review Criteria Required for Acceptance?**

**Methods**

-Are the objectives of the study clearly articulated with a clear testable hypothesis stated?

-Is the study design appropriate to address the stated objectives?

-Is the population clearly described and appropriate for the hypothesis being tested?

-Is the sample size sufficient to ensure adequate power to address the hypothesis being tested?

-Were correct statistical analysis used to support conclusions?

-Are there concerns about ethical or regulatory requirements being met?

Reviewer #1: The manuscript meets all the key criteria

Reviewer #2: Objectives clear and testable

Study appropritate

Population appropriate

No ethical concerns

**Results**

-Does the analysis presented match the analysis plan?

-Are the results clearly and completely presented?

-Are the figures (Tables, Images) of sufficient quality for clarity?

Reviewer #1: The results are clearly presented

Reviewer #2: Analysis matches plan

Results clearly and completely presented

Figs of good quality

**Conclusions**

-Are the conclusions supported by the data presented?

-Are the limitations of analysis clearly described?

-Do the authors discuss how these data can be helpful to advance our understanding of the topic under study?

-Is public health relevance addressed?

Reviewer #1: The conclusions are supported by the data

Reviewer #2: Conclusions supported by data, limitations clearly described

Data well described and contextualised

Public helath relevance addressed

**Editorial and Data Presentation Modifications?**

Reviewer #1: The manuscript contains a significant number of typographical errors but since no page or line numbers are given it is too much effort for this reviewer to have to describe their location. I am happy to do this in a revised manuscript.

Reviewer #2: From the accession number given in the methods section, it appears that the Cry14Ab variant used in the study is Cry14Ab1. It would be helpful if this full designation were given at least once in the manuscript eg in the methods section, if not throughout.

The work involves horse cyathosomins, which is a collection of parasites and may not be commonly known. It might be useful to help the reader by including a short sentence to introduce/describe this term.

In general, the work could be a little more clear about which controls were used in different experiments, and why. 

• The use of two terms, IBa and EVC throughout the manuscript, is a little confusing. If they are the same thing, I would suggest using just one term (or explaining early that they are the same and then using just one term). If IBa is a treated form of EVC, it would be useful for this to be explained clearly at some point and with the reasons for using each as a control in particular circumstances.

• Figure 2 legend, the phrasing “relative to no Cry14Ab alone” is a little confusing (if there’s no Cry14Ab, it can’t be alone). Does this mean relative to control (which control could be specified)? Figures 2 and 3 seem to be compared to a no Cry14Ab control but it is not clear whether this is water, buffer, EVA, IBa.

• Figure 4 seems to use a water control. Why isn’t the control an IBa (or extract from it)?

•In the discussion it is stated that “This report is also the first instance to demonstrate that multiple members of a single family of Cry proteins are anthelmintics”. However, doi 10.1128/AEM.03505-16 indicates antihelminthic activity for Cry5B and Cry5C proteins, so the statement should be removed or modified. 

• Figure 5, I assume that worm burden (panels A and B) is expressed as per animal. Can this be stated (and added to the methods)? Similarly for figure 9

• Figure 6, please state here or elsewhere, which variant of Cry5Ba was used.

• Figure S3: what algorithm was used to identify and assign conserved residues? Different methods will identify different residues as conserved. I am surprised to see at residue 770 for example, the acidic E marked as conservative with the basic K (similarly at 907)

Minor typographical errors: 

• “The doses use in Figure 4 would, a molar level, not ...”: use -> used

• “recognize nematocidal cry”: cry -> Cry

• “Among these, we synthesized, expressed, and found accession no. MF893203 as highly active”. The phrasing here might be improved eg found … synthesised, expressed or change “as” to “to be” depending on the intended meaning

• “5% CO2” , make subscript

• Reference 9 should be corrected to show the publication date as 2021 and to add the volume number (186). References 11, 24, 35, 40, 48 – seem to lack page numbers.

• Figure 9 legend, remove italics from “infections”

• Figure S2 legend “was see” -> was seen

**Summary and General Comments**

Reviewer #1: The manuscript provides convincing data that Cry14Ab and Cry14Ac have activity against GIN parasites. I do though have the following comments that would improve the presentation of the publication (see also comment above about typos).

1) In the third paragraph of the introduction it says that Cry14Ab is related to Cry5Ba - the authors should specify in which way it is related.

2) In the 2nd results section it is stated that the dose of Cry14Ab that gives complete inhibition of H. contortus coreesponds to 380 pM. We are told that in a w/w comparison this is 10x more potent than Cry5Ba so we should be told what the difference is using a molar comparison

3) Although it is probably obvious to those in the field it is not obvious to others why larval stages were used in some assays and adults in others. This should be explained.

4) The section on the identification of Cry14Ac implies that the software used is somehow capable of identifying nematicidal toxin genes. It isn't - it is only capable of identifying genes likely to be related to known pesticidal protein encoding genes. The authors should therefore explain how they narrowed down their search to nematicidals (presumably by searching for genes with homology to those known to have this activity)

5) In many places no gap if left between the text and the in-text citation number

6) Species names are not italicized in the most of the references

7) Use superscript in legend to Fig 1

8) This reviewer struggled to understand the term "no Cry14Ab alone" in the legend to Fig2

Reviewer #2: This is a clear an interesting manuscript describing the activity of two Cry14A variant proteins against a range of nematodes including gastrointestinal parasites of medical and veterinary interest. In vitro experiments are built on with in vivo experiments showing reduced worm burdens or reduced egg production. Overall the manuscript is clearly written and a good contribution to the field with clear implications for future control strategies.

As a general note, it would help the reviewing process if page and line numbers were marked.

PLOS authors have the option to publish the peer review history of their article (what does this mean?). If published, this will include your full peer review and any attached files.

Reviewer #1: No

Reviewer #2: No

Figure Files:

Data Requirements:

Reproducibility:

References

---

## [Decision Letter · Decision Letter 1]

12 Sep 2024

Dear Prof. Aroian,

Thank you very much for submitting your manuscript "Bacillus thuringiensis Cry14A family proteins as novel anthelmintics against gastrointestinal nematode parasites" for consideration at PLOS Neglected Tropical Diseases. As with all papers reviewed by the journal, your manuscript was reviewed by members of the editorial board and by several independent reviewers. The reviewers appreciated the attention to an important topic. Based on the reviews, we are likely to accept this manuscript for publication, providing that you modify the manuscript according to the review recommendations. 

Sincerely,

David Joseph Diemert, M.D.

Academic Editor

Jong-Yil Chai

Section Editor

Reviewer's Responses to Questions

**Key Review Criteria Required for Acceptance?**

**Methods**

-Are the objectives of the study clearly articulated with a clear testable hypothesis stated?

-Is the study design appropriate to address the stated objectives?

-Is the population clearly described and appropriate for the hypothesis being tested?

-Is the sample size sufficient to ensure adequate power to address the hypothesis being tested?

-Were correct statistical analysis used to support conclusions?

-Are there concerns about ethical or regulatory requirements being met?

Reviewer #1: Yes

Reviewer #2: Yes

**Results**

-Does the analysis presented match the analysis plan?

-Are the results clearly and completely presented?

-Are the figures (Tables, Images) of sufficient quality for clarity?

Reviewer #1: Yes

Reviewer #2: Yes

**Conclusions**

-Are the conclusions supported by the data presented?

-Are the limitations of analysis clearly described?

-Do the authors discuss how these data can be helpful to advance our understanding of the topic under study?

-Is public health relevance addressed?

Reviewer #1: Yes

Reviewer #2: Yes

**Editorial and Data Presentation Modifications?**

Reviewer #1: Only minor changes required

Reviewer #2: The issues listed in previous reviewer comments appear to have been addressed satisfactorily

**Summary and General Comments**

Reviewer #1: The authors have addressed my previous concerns. I now only have some minor comments on presentation:

1) Pg 2 Line 12 and throughout. It is no longer standard practice to italicize in vivo and in vitro

2) Pg 6 lines 9-10. Firstly the formatting is a bit odd with the parentheses including citation 26. Secondly are the authors sure that all those toxins that have activity against nematodes cluster together in a phylogenetic tree.

3) Pg7 Line 14 Cry3A

4) Pg9 line 18 "at molar level"

5) Pg10 Line 15 "for detecting...." not "to detecting"

6) Pg13 Line 6 "also a highly"

7) Pg13 Line 7 change to "similar to that published for Cry5Ba [20,24]."

8) Pg14 Line 9 the manuscript is rather overselling the data mining aspect. The only reported outcome of the IDOPS method was a protein with >90% sequence identity to an existing protein. A simple BLAST search would have easily found that! That is not to say that IDOPS is not a useful tool but this manuscript does not provide a strong endoresment.

9) Pg15 Line 15 why is there a superscripted - in the 4D8 spo0A deletion but not in the parent strain? Does this represent the Cry- phenotype?

10) Pg15 Line 23 the gene not the protein was synthesized (lower case italicized).

11) Pg16 Line 4 - see 9) above - yet another version

12) Pg17 Line 13 For the experiment

13) Pg18 Line 12 space after the 2nd 20

14) Pg 20 Line 20 studies was used as intact

15) Pg 22 Line 2 change to "of pH3 solubilized Cry14Ab

16) Legend to Fig 1 yet another way of formating the strain name (see 9) and 11) above)

Reviewer #2: (No Response)

PLOS authors have the option to publish the peer review history of their article (what does this mean?). If published, this will include your full peer review and any attached files.

Reviewer #1: No

Reviewer #2: No

Figure Files:

Data Requirements:

Reproducibility:

References

---

## [Editor Report · Decision Letter 2]

7 Oct 2024

Dear Prof. Aroian,

We are pleased to inform you that your manuscript 'Bacillus thuringiensis Cry14A family proteins as novel anthelmintics against gastrointestinal nematode parasites' has been provisionally accepted for publication in PLOS Neglected Tropical Diseases.

Best regards,

David Joseph Diemert, M.D.

Academic Editor

Jong-Yil Chai

Section Editor

---

## [Editor Report · Acceptance letter]

21 Oct 2024

Dear Prof. Aroian,

We are delighted to inform you that your manuscript, "Bacillus thuringiensis Cry14A family proteins as novel anthelmintics against gastrointestinal nematode parasites," has been formally accepted for publication in PLOS Neglected Tropical Diseases.

Best regards,

Shaden Kamhawi

co-Editor-in-Chief

Paul Brindley

co-Editor-in-Chief
